# Artificial Neural Networks for Flexible Pavement

Ramin Bayat [1], Siamak Talatahari [2,3], Amir H. Gandomi [2,4,*], Mohammadreza Habibi [5] and Babak Aminnejad [6]

1 Department of Civil Engineering, Kish International Branch, Islamic Azad University, Kish Island, Iran
2 Faculty of Engineering & Information Technology, University of Technology Sydney, Sydney, NSW 2007, Australia
3 Department of Civil Engineering, University of Tabriz, Tabriz 5166616471, Iran
4 University Research and Innovation Center (EKIK), Óbuda University, 1034 Budapest, Hungary
5 Department of Civil Engineering, Kermanshah Branch, Islamic Azad University, Kermanshah, Iran
6 Department of Civil Engineering, Roudehen Branch, Islamic Azad University, Roudehen, Iran
* Correspondence: gandomi@uts.edu.au

**Abstract:** Transportation agencies are primarily responsible for building new roads and maintaining current roads. The main focuses of these agencies are to prioritize maintenance and make significant rehabilitation decisions to handle serious problems facing road authorities. Considerable efforts and an abundance of studies have been performed to determine the nature, mechanisms, test methods, and measurement of pavements for preservation and improvements of roadways. The presented study reports a state-of-the-art review on recent advances in the application of artificial intelligence in various steps of flexible pavement, including pavement construction, performance, cost, and maintenance. Herein, the challenges of gathering large amounts of data, parameter optimization, portability, and low-cost data annotating are discussed. According to the findings, it is suggested that greater attention should be paid to integrating multidisciplinary roadway engineering techniques to address existing challenges and opportunities in the future.

**Keywords:** artificial intelligence; flexible pavement; performance; construction; maintenance





## 1. Introduction

Flexible pavement conditions have an enormous impact on road safety and performance and are affected by a variety of factors like traffic patterns, climate, development parameters, construction methodology, and protective methods. Since service life depends on the flexible pavement situation, it is essential to conduct a control procedure as a management system of pavement. To ensure the long-term sustainability of roadways, minimize maintenance costs, and conserve resources, paving materials must be carefully chosen. The optimum design of both the pavement mixture and roadway is of great importance [1].

Artificial Intelligence (AI) is a branch of informatics in which computers perform human-like tasks, such as correctly detecting and learning inputs for perception, problem-solving, reasoning, knowledge representation, and planning. Different innovative AI technologies have been designed to mimic the cognitive capacities of humans to deal with more complex problems intentionally, intelligently, and adaptively. In general, AI may be considered a combination of machine learning and data analysis [2]. Scientists hope to find more precious or attractive information underlying the physical effect of additives and experimental data. AI, as an evidence-based technique with high potential, has had success and dependability in many academic topics and projects. Therefore, this research systematically analyzed the utilization of AI in various types of flexible pavement, including pavement design, construction, cost, and maintenance. Prior studies on this topic over the last decade, especially since 2014, were gathered for this research. Specifically, the analysis aimed to distinguish the modern challenges and future headings of the research, allowing scientists to detect significant areas for subsequent study.

## 2. Methodology

To achieve our research objective, we collected available documentation on the application of ANNs in flexible pavement. This paper covered articles published in the English language between 2014 and 2022, as shown in Figure 1. The key descriptors used in the literature search were created by a combination of various keywords. It was found that the number of articles published has grown rapidly in recent years. The topics discussed in the present paper are summarized in Table 1.

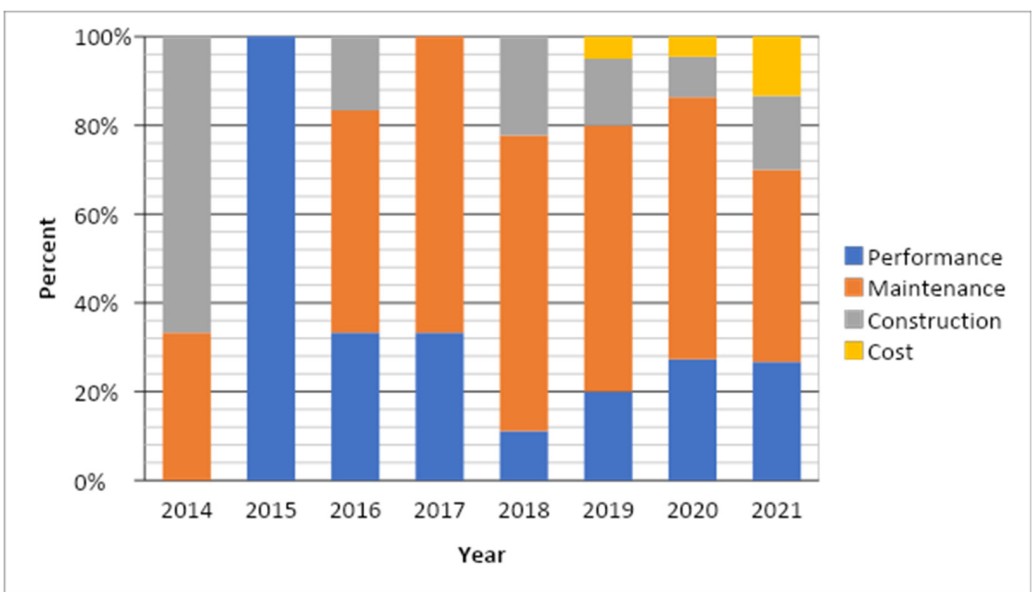

**Figure 1.** Literature search results.

**Table 1.** Key topics of this paper.

| Major Topic | Sub-Topic |
| --- | --- |
| Pavement Performance | Control<br>Classification |
| Pavement Construction | Workability<br>Quality<br>Design |
| Pavement Maintenance | Prediction<br>Planning |
| Pavement Cost | Planning |

The research articles were selected based on the following criteria:

- Hot research topics
- Significant contribution citations
- Publication time
- Contribution of the research methods
- Largeness of utilized data
- Quality of paper

## 3. AI in Flexible Pavement

The flexible pavement design consists of complicated inter-material interactions, vehicle load, road structure, and the environment and requires experienced researchers and complex calculations. As the accessibility of pavement engineering research data has increased, researchers have shifted focus to creating and assessing designs using artificial

intelligence. As a result of the literature review, four research hot topics during the design of the flexible phase were identified: performance, construction, maintenance, and cost.

### 3.1. Flexible Pavement Performance

Pavement performance is a major challenge in road engineering operations and planning. Pavement performance is influenced by of several factors like soil, environment, traffic, and economy, as well as stress distribution factors. The main function of pavement is to provide a smooth driving surface and appropriate surface friction.

Sadat Hosseini et al. [3] found that the constituent viscoelastic characteristics of the complex shear module (G*) and phase angle (δ) allowed for the selection of the optimal dose of an additive for modifying the unique bitumen using crumb rubber, styrene–butadiene–styrene, and polyphosphoric acid. The mixture was optimized using artificial neural networks (ANNs), linear regression, linear help vector regression, choice tree regression, Gaussian system regression (GPR), and ensemble regression. Comparing the different outcomes of the model in terms of the overall efficiency coefficient of the performance measures, the researchers suggested that the overall regression approach yields the best performance in the projections of concrete asphalt.

Behnke et al. [4] produced a qualitative road rutting prediction model to study the long-term structural response of high-stress elastoplastic solids to repeated tire overflows. The model combines a time homogenization technique based on the finite element method with an arbitrary Lagrangian–Eulerian specification. The asphalt mix structural indicators were utilized to develop a virtual design technique for the asphalt microstructure that customizes the morphology and spatial distribution of aggregates.

ANN models can be used to model absorbent asphalt's acoustic properties. Acoustic asphalts are especially useful to reduce noise from road traffic. This solution is ideally suited to urban areas, where noise-absorbing asphalt is utilized to control noise with minimal effect on the environment. In [5], the properties of acoustic asphalt, including the sound absorption coefficient, were found experimentally. Then, the acoustical coefficient of the proposed numerical model and the data of measurements were compared. The final model can be used for predicting the sound absorption coefficient [5].

A prior study developed a suitable model to present the relationships between a material's properties and the acoustic coefficient of absorption to forecast the acoustical characteristics of the material. Via experimentations with acoustic asphalts, the measured sound characteristics of the material were recorded and analyzed. Based on the experimental data, a numerical model was constructed and validated [6].

Amorim et al. [7] reported a multi-layer perceptron neural network (MLPNN) with two concealed layers for determining the ratio of the influence of various configurations of axle loads. Since the existing mechanical-empirical equation is very complicated for the design equivalent single axle load (ESAL) evaluation, an ANN was developed for the computation of EALFs.

Ziyadi and Al-Qadi [8] studied the impact of broad-base tires on flexible pavement using an MLPNN model, containing a k-fold cross-validation method, which can accurately predict the critical responses of the 3D finite element model.

Moussa and Owais [9] developed a dynamic prediction module for hot mix asphalt using a 251-deep convolutional neural network model and six convolution blocks. The convolution block included convolution, batch normalization (BN), and ReLU enabled layers.

Seitllari et al. [10] evaluated variations in the dynamic module of aged asphalt mixture using MLMLNP with a hidden layer. In the meantime, 249 hidden two-layer MLPNN models were used to predict the dynamic module of the hot mix asphalt.

Moghaddam et al. [11] adopted different types of machine learning techniques, likemulti-level factorization net (MLFN), to predict the fatigue life of mixed asphalt modified with polyethylene terephthalate. The results demonstrated the ability of the neural network to predict fatigue lifetime.

Ahmed et al. [12] predicted HMA fatigue life with stress-controlled tests using two MLFN models, namely a strain test model and a strength test model. The strain test model performed better in predicting accuracy.

El-Badawy et al. [13] used an ANN for finding the dynamic modulus (E*) of Witczak NCHRP 1-37A, Witczak NCHRP 1-40D, and Hirsch models. It was found that more precise E* forecasts can be made by using mentioned models. A global sensitivity analysis (GSA) demonstrated that representative parameters of aggregates, binders, and mixtures have converging effects on E* predictions using an applied model.

Majidifard et al. [14] predicted Gf in flexible pavement using gene expression programming (GEP) and ANN/SA methods. The researchers proposed models consisting of various materials like bitumen, aggregate, regenerators and rubber, and recycled materials. While the models provided robust predictions of G, the GEP model outperformed the ANN/SA model for test data. GEP can model Gf without having to pre-define the functional structure of the model. The optimal ANN/SA model transformed into functional representation during calculations. The results showed that the GEP model had better generalization and a simpler functional structure.

Huang et al. [15] used parameters of the Witczak pattern for developing a hybrid algorithm. The dynamic module of the asphalt mixture was predicted using a modified BAS algorithm and RF model. The outcomes of the research showed that the dynamic module is affected mostly by G* and the phase angle of the binders. Although the volumetric properties had some impact, the variation of the variable controlling the aggregation gradation demonstrated little impact on the dynamic module.

Shafabakhsh et al. [16] inspected various aggregates, types of additives, percent of additives, temperature, and stresses to model the deformation of flexible pavement using an ANN model. The ANN training process was done by back-propagation neural networks. The deformation measured the expected deformation by the ANN model and was compared using the coefficient R2. Results showed that the final strain of the asphalt mixture could be modeled with minimal errors and time using the ANN model.

Arifuzzaman et al. [17] investigated the impact of numerous factors, such as environmental conditions, types of binders, and CNT doses, on the behavior of asphalt adhesives. Adhesion force was measured by BOA-based SVR techniques via AI hybrid models. Atomic Force Microscopy was used to estimate the adhesion force of asphalt. The model's performance was evaluated using various performance measurement indicators. Results showed similarities between the mean, median, and typical deviations of the rolling bottle test and model's values. The high efficiency of the developed model was confirmed by low mean absolute error, average square error, and values of fractional bias.

Arifuzzaman et al. [18] developed various models of adhesion properties of NTC and polymers of bitumen binders. Using the corresponding standard method, damage caused by humidity and oxidation was simulated in various samples. The study developed a neural network of the radial base function (RBFNN), including various parameters such as asphalt chemistry, polymer and CNTs type and percentages, and various environmental exposures for predicting the adhesive strength of asphalt to nanoparticles. The adhesive properties of SB-modified asphalt were found to be more consistent than that of nanomodified asphalt.

Vyas et al. [19] surveyed the suitability of using ANN numerical predictive models for structural performance parameters of flexible pavement to optimize the pavement maintenance system. The parameters of functional performance of the paved roadways as well as the environmental and foundation ground characteristics were correlated with the results of the FWD deflection, namely SCI and BCI.

Marcelino et al. [20] proposed a model for predicting pavement performance in restricted data environments. Results show that accurate performance prediction of models can be attained with limited data when a transfer learning approach is implemented. All models derived from this approach outperformed the basic models with regard to long-term forecasts.

Pantusos et al. [21] developed a negative binomial regression model for predicting pavement age-related degradation. The model was compared to traditional nonlinear regression models. Specifically, the linear empirical Bayesian (LEB) approach combines deterioration and measured conditions for improving the predictions. The proposed model can predict the average square error of pavement conditions in the following year based on the measured pavement condition without further modeling of pavement degradation.

Kırbaş and Karaşahin [22] utilized different techniques for predicting pavement deterioration, including deterministic regression analysis, an artificial neural network, and multivariate adaptive regression splines. Results reveal that the ANN model exhibited the greatest accuracy.

Channelized traffic can be reduced by altering lane width and programming a designed lateral wandering and distributing wheel load frequency. However, another important factor that can negatively impact pavement performance due to the automated bogie platoon is the next expected reduced distance. Reducing the next average distance may result in reduced air resistance and, therefore, fuel savings of 2–12% based on the next average distance [23].

Duckworth et al. [24] used ANN models for predicting pavement performance, considering the impact of rehabilitation measures on the inputs like traffic load, climate, and environmental factors. The most promising models were found to be Pavement Condition Rating and International Roughness Index. The ANN model showed to characterize the behavior of pavement, even when the statistical measurements fell outside the appropriate range. Rehabilitation measures were effectively integrated into the model and proved to be accurate.

Issa et al. [25] investigated six of the most commonly encountered pavement defects. Using the Federal Highway Administration Long-Term Pavement Performance database, a hybrid model was proposed for identifying the pavement condition index. To verify the robustness of the model, a non-sampled performance analysis as well as cross-validation analysis were conducted.

Morris et al. [26] proposed a novel model pipeline for detecting pavement wetness based on direct images of road scenes taken by traffic cameras. Two of the most popular gradient enhancement algorithms (XGBoost and CatBoost) were assessed along with a conventional logistic model for the classification task. Based on experimental data with the custom dataset, the CatBoost classifier exhibited the best performance.

Ranjbar et al. [27] provided a brief explanation of computational intelligence (CI) and an overview of CI frameworks. In addition, the methodology for the latest and most efficient techniques in the various CI applications, like data learning, optimizing, and solving problems with uncertainty, are discussed. The authors also gave an overview of CI applications in another portion of the pavement management system (PMS).

A comparison of methods used in the prediction of flexible pavement performance is shown in Table 2.

**Table 2.** Comparison of methods used in prediction of flexible pavement performance.

| Reference | Numerical | Experimental | Methodology |
|---|:---:|:---:|---|
| Sadat Hosseini et al. [3] | ✔ | ✔ | Artificial Neural Networks (ANN), Linear Regression (LR), Gaussian Process Regression (GPR) |
| Behnke et al. [4] | ✔ | | Finite Element Method (FEM) |
| Ciaburro et al. [5] | ✔ | ✔ | Artificial Neural Networks (ANNs) |
| Iannace et al. [6] | ✔ | | Multi-Layer Perceptron Artificial Neural Network (MLPNN) |
| Ziyadi and Al-Qadi, [8] | ✔ | | Multi-Layer Perceptron Artificial Neural Network (MLPNN) |

**Table 2.** *Cont.*

| Reference | Numerical | Experimental | Methodology |
|-----------|:---------:|:------------:|-------------|
| Moussa and Owais, [9] | ✔ | | Convolutional Neural Networks (CNN) |
| Seitllari et al. [10] | ✔ | | Multi-Layer Perceptron Artificial Neural Network (MLPNN) |
| Moghaddam et al. [11] | ✔ | ✔ | Multi-Layer Feed-Forward Neural Network (MLFN) |
| Ahmed et al. [12] | ✔ | | Multi-Layer Feed-Forward Neural Network (MLFN) |
| El-Badawy et al. [13] | ✔ | | Gravitational Search Algorithm (GSA) |
| Majidifard, [14] | ✔ | | Gene Expression Programming (GEP) |
| Huang, [15] | ✔ | ✔ | Beetle Antennae Search (BAS), Random Forest (RF) |
| Shafabakhsh et al. [16] | ✔ | ✔ | Artificial Neural Network (ANN) |
| Arifuzzaman et al. [17] | ✔ | ✔ | Support Vector Regression (SVR) |
| Arifuzzaman et al. [18] | ✔ | ✔ | Radial Basis Function Network (RBFNN) |
| Vyas et al. [19] | ✔ | | Artificial Neural Networks (ANN) |
| Marcelino et al. [20] | ✔ | | Artificial Neural Network (ANN) |
| Pantuso et al. [21] | ✔ | | Linear Empirical Bayesian (LEB) |
| Kırbaş and Karaşahin, [22] | ✔ | | Multivariate Adaptive Regression Splines (MARS) |
| Duckworth et al. [24] | ✔ | | Artificial Neural Networks (ANN) |
| Issa et al. [25] | ✔ | ✔ | Artificial Neural Networks (ANN) |
| Morris, et al. [26] | ✔ | ✔ | XGBoost and CatBoost |
| Ranjbar et al. [27] | ✔ | | computational intelligence (CI) |
| Summary | ANN is the most effective and practical model compared to others in predicting the performance of asphalt pavement and most of the studies conducted were numerical. | | |

### 3.2. Flexible Pavement Maintenance

Different numbers of hidden layers and neurons in various cases of MLPNNs have been developed to anticipate the critical reactions of pavement to the failure of the descending crack. In addition, various architectures and training algorithms of MLPNN have been used to accurately predict the maximum tensile stress exerted by different aircrafts. In general, based on the complexity of the problems, the scale of the neural network should be determined [28,29].

Hussan et al. [30] demonstrated that the performance of rutting is one of the important indicators of a flexible pavement design. Many studies have focused on applying ANNs in predicting rutting performance to eliminate time-consuming tests.

Lau et al. [31] applied the Visual Geometry Group network using an MLPNN for the classification of automated crack, which obtained high precision in publicly available datasets. For improving the CNN classifiers, 3D distress images were utilized to improve the distress function and reduce noise. Afterward, various sensitive field sizes were studied to select the optimal hyperparameters of CNNs.

Huyan et al. [32] detected different sealed/unsealed cracks under complex roadbeds. The influences of off-balance lighting, markings, and shading were considered by developing a CrackDN using a Region-based Convolutional Neural Network (R-CNN) architecture.

Song and Wang [33] applied Faster R-CNN to autonomously find different pavement areas, like cracks, potholes, oil bleeds, and surface points. CNNs based on the region's proposal, even by using Faster R-CNN, could not respond to requests to localize in real time.

Du et al. [34] compared Faster R-CNN to the 'You Only Look Once' (YOLO) model to assess the distress detection performance on the pavement regarding the precise location and processing speed.

Ukhwah et al. [35] reached a compromise accuracy by adopting multiple YOLO architectures with various backend networks, including YOLO v3, tiny YOLO v3, and space pyramid pooling YOLO v3 (SPP). Furthermore, the SSD was also used for the rapid detection of distress on the pavement with little computational effort.

Kang et al. and Liu et al. [36,37] suggested a two-stage segmentation methodology to improve timeliness. The distress segmentation of the pavement was performed within the delineation box generated by the location, thereby reducing computing time in unserved areas. Both works have given a new outlook on the rapid segmentation of distress on the pavement.

FWD and Gaussian System Regression (GPR) are two well-known test devices for the collection of fast, high-frequency data. Following the acquisition of test data, a variety of data mining techniques can be utilized to derive valuable information and evaluate the condition of the pavement's structure. In particular, ANNs have been used for estimating pavement deviations. The retrospective calculation of the pavement module is a critical technique for assessing the condition of the pavement structure using the deflection vessel parameters at the front [38].

GPR has been largely applied to inspect the distress of a pavement structure, such as reflective crack, surfacing of asphalt, and uneven settlement. Many researchers have started to apply deep learning to automatic distress detection due to the difficulty in identifying defects from GPR images. Some studies have extracted time-frequency function vectors from GPR signals through a variety of data processing methods. In addition, recognition algorithms based on MLPNN have been designed for automated detection of conditions like humidity, damage, and monitoring of asphalt road density. Furthermore, a deeper MLFN was used to directly extract the GPR signal feature and then identify any unusual internal defects and signals. Research suggests that CNNs are better suited for GPR image processing [39].

Luca [40] demonstrated that the roughness of flexible pavement is an essential factor for the safety and comfort of vehicles. Many studies established the correlation between IRI and other kinds of distress metrics, such as rutting, cracks, propagation, and pothole (in addition to pavement surface) defects by using MPLNNs. This is pertinent since the condition of a pavement structure impacts IRI.

Fathi et al. [41] analyzed the LTPP database, which includes quality control parameters, such as mineral aggregate voids (VMA), mixture air voids (VA), asphalt concrete field density, pavement structure age, and deterioration indexes. Subsequently, the researchers developed a hybrid Machine Learning (ML) method combining Random Forest (RF) and ANN for Alternating Direction Implicit (ADI) prediction. The results show that the ML hybrid technology is capable of rigorously forecasting road deterioration.

Hafez et al. [42] selected the appropriate maintenance and rehabilitation treatments for the low-volume routes utilizing MLPNN, which covered a range of treatment levels from waterproofing to stacking.

Ziari et al. [43] suggested an ANN forecast approach for short- or long-term IRI values of flexible pavements. Multiple LM-based MLP arrays were utilized for sensitivity analysis. The study concluded that the future state of flexible pavement with satisfactory precision in the short or long term can be predicted by ANN.

Wu et al. [44] used neuronal network models for estimating SIF at fatigue and reflective cracking to remedy the inaccuracy of multivariate regression models and the computational burden of FEA. ANN models showed to be effective enough to achieve results once developed, while their performances have also proven to exceed the non-linear regression counterparts. Six backpropagation ANN models were developed according to the database generated from the FEM.

Yoo and Kim [45] proposed a BP-based MLP 8-12-8-2 model to distinguish between cracks of pavement and sound objects of pavement imagery. The datasets were used for instruction and validation, respectively, and the model yielded very accurate predictive capability.

Alavi et al. [46] demonstrated that Fn is recognized as one of the best predictors of the rutting potential of asphalt blends. The effectiveness of the proposed model was verified through the validation steps. Fn was found to be more influenced by PC than by other mixing properties. The results indicated that the MPMM model significantly outperforms various CI models. In addition, the MGGP model encompasses the effects of most of the necessary parameters to establish an optimal mixing plan.

Hoang [47] established an automated approach for detecting potholes in pavement. Image processing technologies, including GF, SF, and IP, were synergistically used for extracting the characteristics of digital pavement images. Specifically, two GF levels were used as a picture denominator technique, and GF-aided SF was applied for generating a pothole resilience map. The IP analysis based on such a map was carried out to digitally present the characteristics and recognize potholes of an image of special interest.

Mamun et al. [48] conducted some CI techniques for analyzing the damage of moisture in lime and chemically-modified asphalts. Under both dry and wet conditions, statistical methods were developed. Results showed that to capture the complex system of relationships between various chemical functional groups that could affect the adhesion and intermolecular cohesion forces of lime-modified asphalt, chemically-induced moisture damage can be used.

Arifuzzaman [49] reported that flexible pavement failure is a main global concern due to damage caused by moisture. In this research, the causes and forecasting of such damage were studied. The asphaltic binder was altered with carbon nanotubes (CNTs) with very low percentages, then polymer and CNTs were added to the liquid asphalt binder to investigate whether the resultant modified binder improved resistance to moisture damage. An artificial intelligence modeling technique was used for determining the behavior of moisture damage to the modified binder.

Bezerra et al. [50] used DNCNs to distinguish between two classes of discontinuities and U-NET architecture by manually defining objects representing the two classes in some layers of the 3D image. The network was applied to the complete image to distinguish between pores and cracks after optimization (Figure 2). The final network was subsequently applied to pictures of different pellets with good results.

Guo et al. [51] predicted two functional indices, namely the international roughness index (IRI) and RD, considering multiple influencing factors using a comprehensive learning model that deploys a gradient-strengthening decision tree (GBDT). The proposed model can yield more accurate performance values of pavement, provide accurate references for pavement maintenance, and optimize the budgetary availability for highway authorities.

Zhang et al. [52] predicted rutting in paved road segments by combining an empirical mechanistic method with material analysis. Results showed that the expected DR was less than that measured in real-time, but in practice, the measured value was close to the values of previous months.

Choi and Do [53] proposed a model for predicting indicators, such as DR, IRI, and crack rate, using input data derived from Korea's pavement monitoring database based on RNN. The model predicted IRI with good accuracy but produced imperfect results in DR and cracking rates with coefficients of determination below 0.80. Because ANNs

require large amounts of data, they have a better capability for image analysis and distress detection.

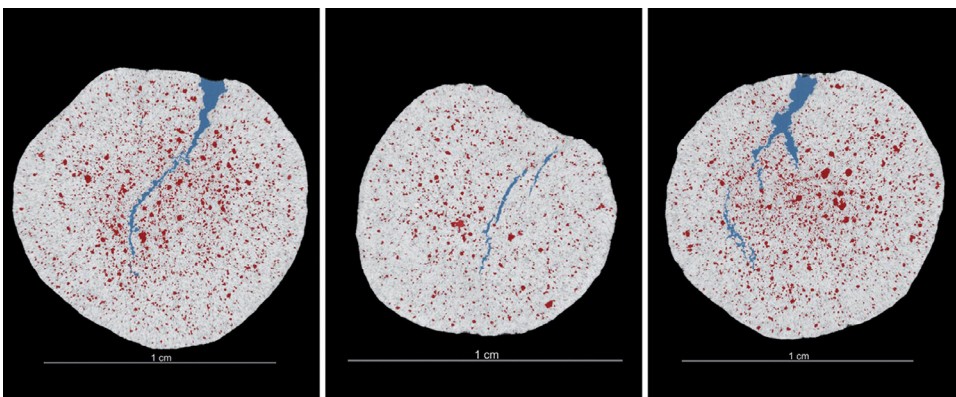

**Figure 2.** Applying a DL network in three different layers of the reference specimen; pores are indicated in red, cracks in blue, and solids in gray adapted with permission from Ref. [50]. 2023, Jório Coelho.

Gong et al. [54] developed a MEPDG model using a set of inputs representing four major influencing criteria to predict rutting: climate, traffic, structure and material properties.

Alatoom et al. [55] used an ANN to develop pavement roughness based on the smartphone measure. IRI values were investigated by considering pavement age, traffic loading, and traffic volume. The results showed that the ANN can anticipate the upcoming IRI with a relatively low average error.

Kouchaki et al. [56] found a strong positive relationship between pavement texture and friction data by analyzing the correlation between the mean depth of the profile and friction data of DFT and GT.

Haddad et al. [57] developed a rutting depth prediction model taking into account the lack of data and resources available in developing countries and local road agencies. Data were extracted from the LTPP database, including a set of climate, traffic, asphalt, base, and subgrade properties. Simplified family models were produced and offered road agencies an extreme lack of resources and a reliable alternative to implementing their pavement management systems.

Pérez-Acebo et al. [58] developed an RFR model to predict the IRI of asphalt pavements by considering 19 different entry characteristics, including environment, traffic, pavement structure, and deterioration. The RFR model exhibited the potential to outperform linear regression methods in training and test packages. However, the cost of calculation time was high because of the large number of trees, confirming the long processing time of traditional GBM and random forest methods.

Liu et al. [59] combined an image-capturing platform, image processing algorithm, and path planning method with a modified 3D printer based on FDM to form an automated pavement crack sealing platform, which is capable of automatically detecting and sealing pavement cracks. The results showed that 3D printing is an efficient technique for the automated sealing of cracks in the pavement.

Liu et al. [60] demonstrated that automated sealing of pavement cracks is a challenging task which has been studied over the past few years, including automated crack detection and AI-based segmentation methods.

Olowosulu et al. [61] used RF and DT algorithms to analyze a comprehensive database from Nigeria's Federal Department of Electricity, Works and Housing, which includes fatigue cracks, average rut depth, and drainage conditions. The results of the analysis showed the performance trend and highway surface condition classification according to surface constraints. The RF and DT algorithms achieved a more precise classification of constraints compared to the NB algorithm.

Furthermore, the use of experience-based models that can meet the challenges of missing data in a dataset has been suggested for developing an optimal PMS in Nigeria.

Aleadelat et al. [62] examined the ability of an inexpensive depth camera (D435i) to measure IRI as pavement indicator rugosity. The D435i depth camera was able to estimate the IRI of asphalt roads with reasonable certitude relative to a standard profiler. The statistical analysis revealed no differences between the two measurement methods.

Elwardany et al. [63] developed a reliable multivariate regression and ANN model to predict fatigue life based on mixture parameters, test temperature, applied deformation, and rest period. The ANN model was used to estimate the impact of a truck platoon on the fatigue life of the pavement as a result of the anticipated reduction in distance and subsequent rest periods. The platoon fatigue coefficient (PFLR) was affected by temperature, applied stress level, and mixing parameters. The significant factors for PFLR also include the level of applied deformation and the binder content.

Duo Ma et al. [64] developed a method for detecting pavement cracks based on a CNN comprising several characteristic layers. The model extracts multiscale characteristics to raise the precision of pavement crack recognition. The results demonstrated that the model could be used to identify cracks in real-time using multi-scale ratio anchorage boxes and multi-scale characteristic maps.

Ghanizadeh et al. [65] analyzed the flexible structure of pavement and identified its critical responses under the effect of a standard axle load with an ANN. In order to analyze the pavement section, multilayered elastic analysis theory and critical pavement responses were considered. Results showed that the ANN has many benefits, such as reducing analysis time, predicting fatigue and rutting lives, and the optimal design of pavement structure.

Omranain et al. [66] used the Superpave mix model, ANN, and Supporting Vector Machine (SVM) techniques to quantify the impact of aging on the rutting of asphalt mixture behavior. Lengthening the aging period reduced the cumulative strain, while increasing the temperature raised the relevant value of all the analyzed samples. The results showed that ANN was the best technique for predicting the impacts of aging on the rutting of flexible pavement.

Solatifar et al. [67] extracted DTCP data to develop a pavement deterioration model using BPNN. The results suggest that the BPNN model could predict pavement roughness degradation with very high precision and less error compared to the polynomial regression model.

Domitrovic et al. [68] evaluated the condition of existing roads and its possible application to define the strategy for maintaining national roads using ANNs. A neuronal network of retro-propagation was applied to 481.3 km of national roads in Osijek-Baranja County. Results demonstrated that an ANN was useful to maintain pavement and strategies of rehabilitation and also for assessing road conditions at the project and network levels.

Inkoom et al. [69] used a step-by-step introduction of partition, bootstrap forest, gradient-doped trees, nearby K-neighbours, naive Bayes, and traditional multivariate linear regression techniques. In order to evaluate each model's stability and robustness, predictive accuracy, relative differences, and the level of predictive precision were estimated.

Yu et al. [70] carried out a study for developing deterioration patterns based on preface findings from a typical successful maintenance survey. A long-term performance database for semi-rigid asphalt pavement was evaluated using a large quantity of survey data and associated environmental parameters by a combination of engineering practices. The performance degradation models for the semi-stiff asphalt pavement were constructed using the statistical regression method. The deterioration models were practical to determine the maintenance decision for semi-rigid asphalt pavements.

Hoang et al. [71] developed a model to classify cracks of flexible pavement. In order to extract functions, image processing techniques, including steerable filters, the projective integral of the image, and an enhanced method of image thresholding were used. As a

result, the proposed automatic approach may be of assistance to transport agencies and inspectors for assessing the condition of roadways.

Shuai et al. [72] introduced a new methodology for detecting and segmenting pavement cracks based on 2D images. In particular, an adjustable filter was employed to improve the contrast of the cracks and the surrounding pavement and capture fracture discontinuity and curvature. Analysis of the crack fouling chart revealed a region of the coarse crack and approximate estimates of crack properties. The coarse cracking zone was introduced into a working contour model, and a step-by-step method was implemented for the crack segmentation model.

Wang et al. [73] investigated the issues of inadequate maintenance procedures, maintenance times, and excessive use of funds to maintain highway asphalt pavements in China. A pavement maintenance and prediction model was developed for preventative road maintenance using a neural network. The results demonstrated that the predictive model conforms to the trend of the development of measured results. The study is of critical importance to future road management.

Han et al. [74] offered an automated vision detection method of pavement cracks with deep learning technology, where a CNN was developed to learn crack characteristics from images with no pre-treatment. The CNN design was prepared using a database of images based on the open-source frame TensorFlow by the Google Brain team and data with high accuracy. Test results showed that the performance of the proposed methodology is satisfactory and, therefore, could be useful to offer an alternative to automated pavement crack detection.

Kumar et al. [75] trained and tested an ANN model using various algorithms, including LM, BR, and SCG algorithms. Stress data on flexible pavements collected from field surveys were used to develop and test the ANN model. The LM algorithm outperformed the BR and SCG algorithms, while the ANN model demonstrated greater accuracy than SVM and RF models.

Kim et al. [76] developed an ANN model to predict the indirect traction resistance (STI) of the middle layer of all sections of paved asphalt on a freeway, using IRI, surface distress, rut depth, and equivalent single axle load as variables. ITS was predicted by a transmission process ahead of the training stage. The model was validated by analyzing the correlations among the planned TSIs based on data from the training and test systems. Lastly, the model was supplemented with the respective min and max TSIs measured in the target section.

Naseri et al. [77] selected the strongest evolutional and metaheuristic algorithms to solve the M&R planning optimization problem, such as WCA, AOA, DE, ACO, GA, and PSO. After comparing the algorithms, WCA and AOA demonstrated the highest performance.

A comparison of the mentioned models and goals in asphalt mixture maintenance is summarized in Table 3.

**Table 3.** Comparison of models and goals in asphalt mixture maintenance.

| Reference | Model | Goal | Finding |
|-----------|-------|------|---------|
| Rezaei-Tarahomi et al. [28] | MLPNN | Crack | Computing the critical stress responses is associated with top-down cracking in multiple-slab rigid airfield pavements. |
| Tarahomi et al. [29] | MLPNN | Tensile stress | Top-down critical tensile stress sensitivity was determined similarly to the 3D-FE model. |
| Hussan et al. [30] | ANN | Rutting | High prediction performances of artificial neural network (ANN) modelling technique was compared to nonlinear regression modelling technique. |
| Lau et al. [31] | MLPNN | Crack | Deep learning technique could solve pavement crack segmentation tasks accurately. |

**Table 3.** *Cont.*

| Reference | Model | Goal | Finding |
|---|---|---|---|
| Huyan et al. [32] | R-CNN | Crack | The performance of sealed crack detection is better than unsealed crack detection for most background conditions. |
| Song and Wang, [33] | R-CNN | Crack, Pothole | Comparing the CNN and K-value method, the optimal Faster R-CNN located pavement distresses with bounding boxes more precisely. |
| Du et al. [34] | R-CNN, YOLO | Distress | OLO-based approach was able to detect PD with high accuracy, which requires no manual feature extraction and calculation during detecting. |
| Ukhwah et al. [35] | YOLO | Distress | YOLO technique had a high opportunity to be developed and implemented as a tool for road assessment. |
| Kang et al. [36] | ANN | Distress | The modified DTM algorithm provided high accuracy with respect to crack length. |
| Mousa et al. [38] | ANN | Deflection | Bonding index varied with the characteristics of the base layer. Non-stabilized base layers experienced relatively weak interface bonding at the AC/base interface. |
| Gao et al. [39] | MVA, ANN | Distress | Through these two models, by simply knowing the IRI, it was possible to indirectly evaluate the "Bearing Capacity" at any point of the runway |
| Luca, [40] | MPLNN | IRI | Through these two models, by simply knowing the IRI, it was possible to indirectly evaluate the "Bearing Capacity" in any point of the runway |
| Fathi et al. [41] | ML, ANN | Distress | The hybrid ML technique was capable of predicting pavement deterioration rigorously. |
| Hafez et al. [42] | MLPNN | Distress | The implementation gaps of pavement-preservation activities among CDOT regions result from limited maintenance funding. |
| Ziari et al. [43] | ANN | IRI | ANN models predict future conditions of pavement with high accuracy in the short and long terms; GMDH models do not have accepted accuracy. |
| Wu et al. [44] | ANN | Fatigue | Advantage of ANN over multivariable regression on the prediction accuracy. |
| Yoo and Kim, [45] | MLP | Crack | An intelligent algorithm was developed which can distinguish crack and noise by eliminating the noise. |
| Alavi, [46] | MGGP | Rutting | The MGGP model performs superiorly to the models found in the literature. |
| Hoang, [47] | ANN | Pothole | The proposed AI approach used with LS-SVM has high potential to assist transportation agencies and road inspectors in the task of pavement pothole detection. |
| Hassan et al. [48] | ANN, SVR | Moisture damage | The ensemble of CI along with statistical measurement provide better accuracy than any of the individual CI techniques. |
| Arifuzzaman, [49] | ANN | Moisture damage | Multi-Layer Perceptron (MLP) provides the best prediction for wet and dry samples. |
| Bezerra et al. [50] | DCNN | Crack | The network was applied to the full image, successfully discriminating between pores and cracks. |
| Guo et al. [51] | GBDT | IRI | The proposed model can provide more precise pavement performance values and may be useful for providing accurate reference for pavement maintenance. |
| Zhang et al. [52] | ANN | Rutting | The combination of PME and MEA proves to be appropriate to evaluate rutting potential in project level pavements. |

**Table 3.** *Cont.*

| Reference | Model | Goal | Finding |
|---|---|---|---|
| Choi and Do, [53] | RNN | IRI | The life cycle of road pavement can be optimized by increasing its life expectancy and reducing its maintenance budget. |
| Gong et al. [54] | RFR | IRI | Both of the developed NNs, particularly the NN20, exhibited significantly better predictive performance than the two MLR models. |
| Alatoom et al. [55] | ANN | IRI | ANN models are more accurate in IRI prediction than the regression models. |
| Kouchaki et al. [56] | DFT | Friction | The developed LLS prototype was able to scan the pavement surface texture more reliably and precisely than the CTM in terms of vertical and horizontal resolution. |
| Haddad et al. [57] | DNN | Rutting | Generic family rutting predictive curves corresponding to specific traffic, climate, and performance combinations were developed to render rutting predictions available to all road agencies. |
| Liu et al. [59] | FDM | Crack | 3D printing is an effective method for automated pavement crack sealing, which is recommended in the field of automatic road maintenance and repair. |
| Liu et al. [60] | ANN | Crack | The precision, recall, and F1 score of the proposed method are higher than other state-of-the-art pavement crack detection methods. |
| Olowosulu et al. [61] | RF, DT | Distress | The RF and DT algorithms yielded more accurate classification compared to the NB algorithm, which could not handle instances of missing data efficiently. |
| Aleadelat et al. [62] | ANN | IRI | The proposed approach has the potential to be a baseline for an inexpensive data collection system suitable for local agencies. |
| Elwardany et al. [63] | ANN | Fatigue | The Platooning Fatigue Life Ratio (PFLR) was found to be dependent on temperature, applied strain level, and mixture parameters. |
| Duo Ma et al. [64] | CNN | Crack | The model was optimal in terms of F1 score and precision-recall curve, was less affected by shadows and road markings, and detected the crack boundaries more accurately. |
| Ghanizadeh et al. [65] | ANN | Fatigue | Application of artificial neural networks for pavement analysis reduces the analysis time and can be used as a quick tool for predicting fatigue and rutting lives of different pavement sections. |
| Omranain et al. [66] | ANN, SVM | Aging | The developed model can be embraced by the pavement management sector for a more precise estimation of the pavement life cycle. |
| Solatifar et al. [67] | BPNN | IRI | Results revealed that predicted IRI values with the developed ANN model have a good correlation with measured values rather than the polynomial regression model for both GPS-1 and GPS-2 sections. |
| DOMITROVIĆ et al. [68] | ANN | Rehabilitation strategies | Artificial neural networks could be used for the optimization of maintenance or rehabilitation strategies and for the assessment of pavement condition at the project and network level. |
| Inkoom et al. [69] | MLRT | Crack | The machine learning methodologies were promising in predicting the crack of pavement based on the $R2$ statistics. |

**Table 3.** *Cont.*

| Reference | Model | Goal | Finding |
|-----------|-------|------|---------|
| Yu et al. [70] | RF, DT | Distress | The proposed deterioration models were useful and practical for the establishment of the maintenance decision of the semi-rigid asphalt pavements. |
| Hoang et al. [71] | SVM | Distress | The proposed automatic approach can assist transportation agencies and inspectors in the task of pavement condition assessment. |
| Yu et al. [72] | ANN | Crack | The estimated crack properties provide information to automatically adjust the parameters of the active contour model for effective and efficient crack segmentation. |
| Han et al. [74] | CNN | Crack | Results show the prospects and potential limitations of DL-based methods in SHM applications. |
| Kumar et al. [75] | SCG, BR | Distress | The ANN model is capable of predicting the PCI with a high level of reliability. |
| Kim et al. [76] | ANN | IRI | An artificial neural network model was developed for predicting the indirect tensile strength (ITS) of the intermediate layer of all asphalt pavement sections in an expressway. |
| Naseri et al. [77] | WCA, AOA, DE, ACO, GA, PSO | Distress | Compared to AOA, DE, ACO, PSO, and GA, WCA's objective function was calculated to be 45%, 74%, 74%, 77%, and 83% less, while its M&R cost was cheaper by 13%, 16%, 27%, 19%, and 18%, respectively. |
| Summary | | | ANN model and predicting crack in asphalt pavement have been the most effective and practical models in flexible pavement maintenance field. |

### 3.3. Flexible Pavement Construction

Mallick [78] introduced a framework using a computational approach based on artificial intelligence (AI) to accelerate a pavement combination design. It was found that the optimum values were obtained by combining various tests, and the test properties may be strongly interrelated. Precise machine learning models can be developed using the test database.

Androjić and Dolaček-Alduk [79] examined the influence of different types of asphalt mixtures, humidity content, hourly capacity, and production temperature to anticipate the natural gas consumption in the process of producing HMA using MLNPN. Meanwhile, aggregate temperature showed to be a vital factor affecting the consumption of energy in HMA generation.

Abed et al. [80] proposed an ANN model to predict the impact of deflection, temperature, and additives on the stiffness of HMA. Numerous variables, lab test conditions, and data acquisition were combined to develop the model. The results revealed that the ANN model can effectively estimate the stiffness of HMA as well as a good relationship between the actual and expected values and a determination factor.

Specht et al. [81] presented results for measuring and simulating viscosity in AR binders. To reduce the cost and duration of the experiments, various combinations of parameters were chosen from a statistical design scheme. For predicting the involved parameters and optimizing the AR binders, MR analysis, ANN, and fuzzy logic were used.

Ozturk and Kutay [82] introduced an ANN model to predict Superpave asphalt mixture design properties like the percentage of VTM at various levels of rotation and maximum density. The dataset used consists of a large number of randomly selected mixing schemes for training. The most precise ANN design was determined to be LM-based MLP 18-300-300-600-4 after 800 tests. The model was capable of producing good results and reducing the design process by a minimum of 3–6 days.

Leiva-Villacorta et al. [83] developed ANN models that can reliably predict pavement seam modules. A database was also created using a stratified-elastic analysis for a flexible three-ply structure of pavement. A total of 100,000 data points were generated per ANN. The most appropriate model was found to be a BP-based MLP 13-20-20-3 network, which achieved highly correlated estimates.

Sebaaly [84] developed an AMO model based on ANN and GA to automate the selection of aggregate gradation and binding grade in the design of an asphalt mix, using an advanced neural propagation network with a single hidden MLP layer. Implementation of the approach was successful on the Marshall blends. Comparing the ANN predictions with the results measured in the laboratory shows that the ANN can predict Marshall properties within the ASTM accuracy range.

Fadhil et al. [85] introduced an ANN model as a HMA design tool based on actual data for the work composition formula. The model showed to reduce the cost, time, and data for designing the mixture, exhibiting high correlation based on a high-quality analysis, and thus could be used as a design tool for HMA.

Zou et al. [86] proposed a neuronal network model to predict SFC as a function of 3D pavement microtexture and macrotexture parameters. The experiment contained 60 pairs of friction and pavement texture data. A porTable 3D laser scanner and a digital sand tester were used to collect information on the pavement texture, while a SCRIM was applied to acquire SFC data on the pavement. 3D texture data were then decomposed into microtexture and macrotexture. Next, the TD values of the digital sand tester and the height, functionality, and hybrid parameters of the 3D microtexture and macrotexture were prepared to characterize the pavement texture.

Enríquez-León et al. [87] demonstrated that the most widely used segmentation technique, namely the manual selection of the threshold (TH), depends considerably on the competencies of the operator, homogeneity of the image, and material complexity. These factors may restrict the reproducibility of the TH method. As part of the study, images of a sample of flexible pavement were acquired from a modern high-resolution microscanner to identify its audiovisual content using various segmentation tools.

Mohamed Jaafar [88] focused on modeling the progress of deteriorating asphalt pavement conditions and performed computer simulations of un-grooved and cracked flexible pavement subsurface models. The research achieved three aims: to evaluate and improve predictive models of deteriorating pavement conditions; to evaluate retrospective design methods of modules for characterization of layers of selected test sections; and to inspect the impact of cracking on pavement responses and implementation of pavement condition decay models to improve the structural design and management of flexible pavements.

Deng et al. [89] suggested a method to obtain flexible pavement layer modules, including a power function describing the AC layer module gradient at various frequencies of loading. Fast Fourier transform, finite element model updating, the kriging model, and artificial intelligence were used. Layer modules of the proposed method were compared with other retrospective computing software for validation.

Soloviev et al. [90] introduced a deep CNN model for identifying defects in road surface imagery. The model was implemented as a simplified and optimized version of the most popular networks that are currently completely connected. Training and learning techniques of the network were performed in two stages, according to the specificity of the problem to be solved. The work demonstrates that such architectures can be successfully constructed by using a small amount of initial data on the conditions.

Georgiou et al. [91] studied ANN and SVM to predict the IRI based on several years of data collection. The results showed that both models can predict the index and are useful in pavement engineering strategies.

Amãndio et al. [92] studied a gap in the literature regarding the sustainability and earnings associated with the implementation of multipurpose optimization in the planning of projects to rehabilitate pavements (Figure 3). The input is associated with the development of a system that can support decision-making throughout the design phases.

The results show the feasibility of the system to rapidly produce optimal solutions that support decision-making and improve flexibilities and efficiencies from the perspective of decision-makers.

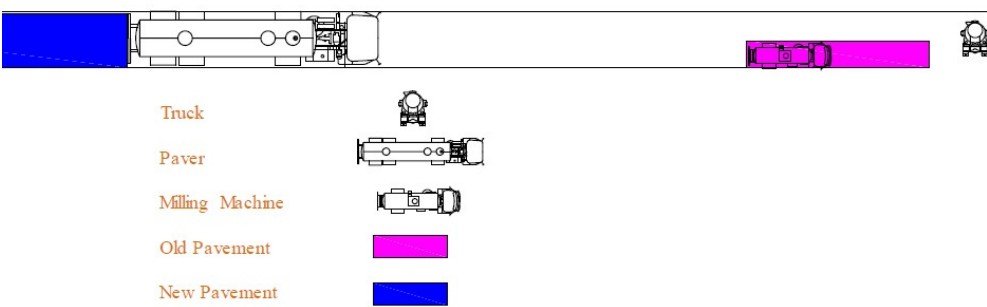

**Figure 3.** Pavement rehabilitation production line [92].

Table 4 presents a comparison of goals in asphalt mixture construction based on the literature.

**Table 4.** Comparison of goals in asphalt mixture construction.

| Reference | Goal | Additive |
|---|---|---|
| Mallick [78] | Optimization of design | - |
| Androjić and Dolaček-Alduk [79] | Natural gas consumption in HMA | - |
| Abed et al. [80] | Optimization of construction | ✔ |
| Specht et al. [81] | Optimization of construction | ✔ |
| Ozturk and Kutay [82] | Design properties | - |
| Leiva-Villacorta et al. [83] | Predicting pavement layer moduli | - |
| Sebaaly [84] | Predict aggregate gradation | - |
| Fadhil et al. [85] | Reducing design time | - |
| Zou et al. [86] | Pavement SFF | - |
| Enríquez-León et al. [87] | AV content | - |
| Deng et al. [89] | Layer moduli | - |
| Amˆandio et al. [92] | Pavement rehabilitation production | - |

*3.4. Flexible Pavement Cost*

Newstead et al. [93] built a test section with various pavement materials to assess and understand how preservation methods differ from traditional methods of flexible pavement. Before construction, the current conditions of the pavement were considered, including stone mastic asphalt (SMA) and high traffic asphalt (HTA), at specified locations along a 1.5 km stretch of urban arterial roadway. The costs, timelines, and placement of materials were evaluated. It was found that SMA and HT may increase the lifespan of the pavement.

Han et al. [94] proposed a smart decision-making model for the maintenance of pavement plans using algorithms to optimize proximal policies to meet the increasing demand for the maintenance and cost of roadways. The decision-making model considers the full maintenance cost-benefit ratio throughout the road's lifecycle, leading to decision-making between the pavement condition and data-based maintenance plans. This incorporates extraction technology to overcome the issues of manual decision-making based on experience. Furthermore, a method to construct an enhanced learning environment module based on a network of deep artificial neurons was proposed, and a reward feature was designed for highway maintenance decisions.

Nahoujy [95] proposed a new ANN approach to calculate deviations at any arbitrary point along a measured roadway using FWD to supplement and replace experimental measurements. The model was developed based on backpropagation by a multilayer perception network for asphalt pavement. This method offers great potential for optimizing traditional measurements in terms of measurement costs and can significantly improve the precision of route maintenance planning by addressing the problem of limited or missing datasets.

Gomes et al. [96] presented and applied different ANN architectures to roadway distress detection. In a supervised environment, Variational Autoencoder (VAE) offers an excellent distinction between good and bad pavement. The developed ANN models can be used as an alternative solution for reducing operating costs relative to expensive business systems and to improve the user-friendliness of traditional road surface classification methods.

Tohidi et al. [97] compared the efficiencies of the genetic algorithm and PSO in the determination of economically-optimum pavement depth. Using algorithms for pavement design, a simulation–optimization model was developed. The results showed that the use of GA and PSO reduced costs of design relative to manual design and that GA outperformed PSO in terms of cost savings.

Fani et al. [98] proposed a stochastic model, considering annual budget and pavement deterioration rate as uncertain pavement factors. They found that the complexity of the model increased as the number of network sections and scenarios increased. The Progressive Hedging Algorithm was used to reduce the cost of maintenance. Compared with a deterministic model, the proposed model demonstrated better results in the field of maintenance and rehabilitation.

## 4. Discussion

A significant increase in research activity in recent years can be observed in Figure 1. The extracted literature is classified by the origin country of the establishment of the first declared author, as shown in Figure 4. A general overall research distribution is discernable, and the chart shows that the United States hosts most of the establishments of the authors. Further, Figure 5 displays the extracted literature classified according to the publisher, revealing Taylor & Francis at the forefront of the field.

This research systematically analyzed the utilization of AI in various types of flexible pavement, such as pavement design, construction, cost, and maintenance.

Table 5 compares some of the models in various references.

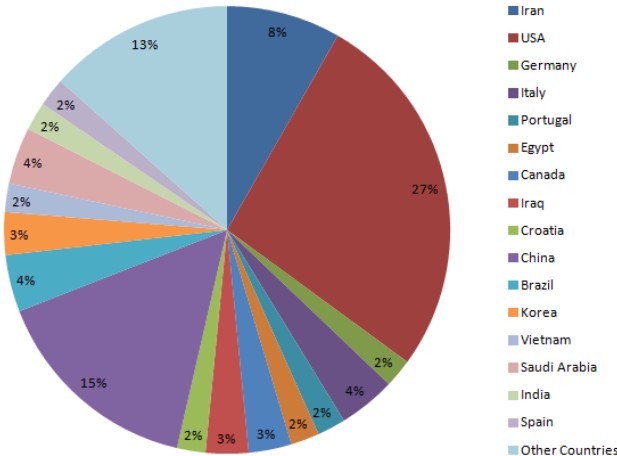

**Figure 4.** Chart of countries of the first author of the extracted research.

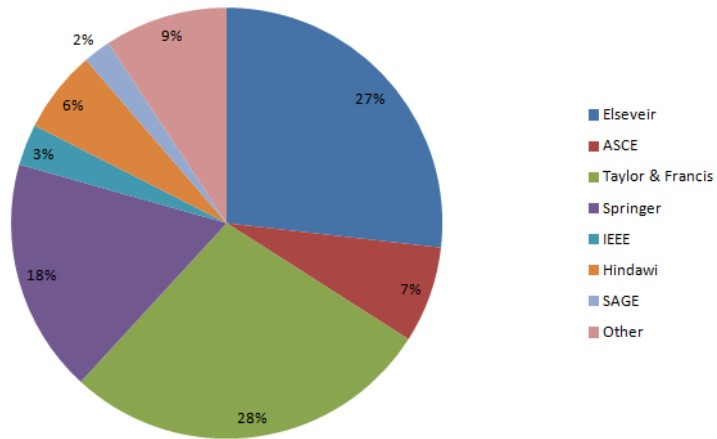

**Figure 5.** Chart of publishers of the extracted research.

**Table 5.** Comparison of models.

| Problem | Model | R2 | RMSE | Best Result |
|---|---|---|---|---|
| Viscoelastic Behavior | ANN | 1 | 0.89 | |
| | SVR | 0.9 | 0.34 | |
| | DT | 0.98 | 0088 | ER |
| | GPR | 1 | 0.34 | |
| | ER | 0.99 | 0.0031 | |
| Flow Number | Model | R2 | MSE | Best Result |
| | MGGP | 0.94 | 1088 | |
| | GEP | 0.77 | 4573 | MGGP |
| | MEP | 0.89 | 2137 | |
| | GP | 0.89 | 2121 | |
| Moisture Damage | Model | NRMSE | MAPE | Best Result |
| | SVR | 0.61 | 0.16 | |
| | ANN | 0.6 | 0.15 | ANN |
| | ANFIS | 0.69 | 0.25 | |
| Asphalt Performance | Model | R2 | MSE | Best Result |
| | ANN | 0.81 | 0.054 | ANN |
| | MLR | 0.47 | 0.032 | |
| IRI | Model | R2 | RMSE | Best Result |
| | ANN | 0.84 | 0.25 | |
| | RFR | 0.88 | 0.21 | GBM |
| | GBM | 0.9 | .19 | |
| Crack | Model | R | F1 Score | Best Result |
| | YOLO | 94.5 | 0.88 | RCNN |
| | RCNN | 96.5 | 0.92 | |
| Pavement Deterioration | Model | MSE | RMSE | Best Result |
| | ANN | 0.04 | 0.07 | ANN |
| | Polynomial | 0. 275 | 0.275 | |

## 5. Conclusions

In this paper, a systematic review of AI applications in flexible pavement was performed with a particular focus on advanced research at different stages of pavement engineering. More than 200 research papers were extracted from databases of digital scientific literature and reduced to 90+ papers relevant to the topic. Quantitative and qualitative analyses reveal an overall introduction of AI applications, which have been particularly significant in recent years. Based on the analyses, the following research fields, along with the topics discussed most in each field, were identified:

- Maintenance field, cracking
- Cost field, budget
- Construction field, design parameters
- Performance field, deformation

This study presents an efficient approach for the application of neural networks in pavement engineering. Pavement performance modeling continued to be a major research area since local calibration of mechanical empirical models must be carried out, after having switched from mechanical modeling. Although data collection became almost fully automated, there were technological implementations, especially in cracking classifying methods and tools.

From the above analysis, it can be concluded that the proposed neural network is appropriate for the classification of data on pavement conditions to determine the global performance index and optimal maintenance strategy. The implementation of AI in a pavement management system would provide a high-quality tool that will facilitate decision-making in the selection of maintenance procedures and the rehabilitation of pavement in individual sections.

The authors suggest a more direct interaction between experts in pavement to further improve pavement engineering. For instance, mathematicians should aim to provide optimization tools that can be easily used by non-experts. The intention of this report was to analyze and summarize all the efforts that have been made in the field of flexible pavement. The following aims are suggested for future works:

- Analyze and compare the effect of various additives in flexible pavement in the construction field.
- Inspect and compare various distresses in the maintenance field using AI.
- Combine and compare experimental and numerical in the performance field.
- Inspect the costs of design, construction, performance, and maintenance process in the flexible pavement field.

**Author Contributions:** Conceptualization, R.B. and S.T.; methodology, R.B.; software, R.B., S.T., A.H.G., M.H. and B.A.; validation, R.B.; formal analysis, R.B.; investigation, R.B., S.T., A.H.G., M.H. and B.A.; resources, R.B.; data curation, R.B.; writing—original draft preparation, R.B., S.T., A.H.G., M.H. and B.A.; writing—review and editing, R.B., S.T., A.H.G., M.H. and B.A.; visualization, R.B. and S.T.; supervision, S.T. and A.H.G.; project administration, S.T. All authors have read and agreed to the published version of the manuscript.

**Funding:** This research received no external funding.

**Data Availability Statement:** Not available.

**Conflicts of Interest:** The authors declare that they have no conflict of interest.

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
