# Peer review of "Artificial Neural Networks for Flexible Pavement"

_information, doi:10.3390/info14020062_

Round 1
Reviewer 1 Report
The article is interesting and informative.
Author Response
The authors would like to thank the area editor and the reviewers for their precious time and invaluable comments. We have carefully addressed all the comments. The corresponding changes and refinements made in the revised paper

Reviewer 2 Report
Authors put huge effort in this review paper. Paper is covering wide area of use of Artificial Neural Networks in pavement applications; consequently, no wonder there are many mistakes and unclear parts of text:
- First question is why time period from 2014 and 2021. Also before that period papers with this topic were written.
- Numbers of figures are completely mixed. There is only once written fig3, in other cases is written Figure.
- Tables 2, 3 and 4 are not mentioned in the text.
- References in text have wrong numbers after reference number 35. Numbers of references 12 and 14 are written twice. Names and surnames of authors are written in different styles: full names, only initials, name before surname, name and surname written together without space between them. References must be unified and take care about spaces.
- In the text many times abbreviations are used without explanations: PMS MARS… Especially GPR is abbreviation for test method and methodology. Especially in tables it would be useful to write full names of methodology.
- For some references we noticed that title of paper is different than description of work in the text: r Gong (53), Perez-Acebo (57), Georgiou (91).
- DOMITROVIĆ is written with capital letters in the text.
Generally paper is well written, but there are many unclear topics and formal mistakes. The time period of this review should be justified- maybe with additional information about when the first application of NNs in this field was observed and how the frequency was changing (growing)?
Author Response
The authors would like to thank the area editor and the reviewers for their precious time and invaluable comments. We have carefully addressed all the comments. The corresponding changes and refinements made in the revised paper are summarized in our response below:
First question is why time period from 2014 and 2021. Also before that period papers with this topic were written.Before that period some period some papers are written and also the writers wanted to show new finding
We thank the reviewer for precious observations. We tried to collect newest papers and some papers are written about this title before 2014.
Numbers of figures are completely mixed. There is only once written fig3, in other cases is written Figure.
Numbers of figures from 1 to 6 were corrected in “Fig” format.
Tables 2, 3 and 4 are not mentioned in the text.
Tables 2,3 and 4 are mentioned in the pages 5,13 and 20 respectively.
References in text have wrong numbers after reference number 35. Numbers of references 12 and 14 are written twice. Names and surnames of authors are written in different styles: full names, only initials, name before surname, name and surname written together without space between them. References must be unified and take care about spaces.
Format of references and numbers were checked and corrected according to reviewer’s valuable points.
In the text many times abbreviations are used without explanations: PMS MARS… Especially GPR is abbreviation for test method and methodology. Especially in tables it would be useful to write full names of methodology.
Abbreviations were added in pages 5,8 and also completely in Table 2.
For some references we noticed that title of paper is different than description of work in the text: r Gong (53), Perez-Acebo (57), Georgiou (91).
Description of the work of Gong in page 10 and Perez in page 11 were moved and they were corrected.
The text of Georgiou in page 19 was also corrected according to reviewer’s comments.
DOMITROVIĆ is written with capital letters in the text
The word was corrected as “Domitrovic” in page 12
Generally paper is well written, but there are many unclear topics and formal mistakes. The time period of this review should be justified- maybe with additional information about when the first application of NNs in this field was observed and how the frequency was changing (growing)?
The first application and basis of NN was in the previous paper that the dear editor advised to omit it.

Reviewer 3 Report
This work provides a review for the machine learning methods for flexible pavement. Four sub areas are targeted, i.e., construction, performance, cost, and maintenance. The introduction is reletively short, which could not convey sufficient findings and contributions for the work. Also, for methodology, it is hard to capture the qualitative metrics regarding the definition of the criteria. This is critical. In the meantime, what do you mean by English Language between 2014 and 2021? Is it from Google Scholar? Or IEEE? Or DBLP etc? The message should clear. Other challenges will be on each section for the four sub areas. It is hard to know, the exact contribution, pros and cons for each work. What is the benefit of knowing their methods being numerical, or experimental, or taking any specific models? What would be the nexgt step?
Thus, it will be difficult to define this work as a systematic review, without providing the significant information, and discussing the critical contributions and comparison for the selected work. Please clearly define it.
Author Response
The authors would like to thank the area editor and the reviewers for their precious time and invaluable comments. We have carefully addressed all the comments. The corresponding changes and refinements made in the revised paper are summarized in our response below:
This work provides a review for the machine learning methods for flexible pavement. Four sub areas are targeted, i.e., construction, performance, cost, and maintenance. The introduction is reletively short, which could not convey sufficient findings and contributions for the work. Also, for methodology, it is hard to capture the qualitative metrics regarding the definition of the criteria. This is critical. In the meantime, what do you mean by English Language between 2014 and 2021? Is it from Google Scholar? Or IEEE? Or DBLP etc? The message should clear. Other challenges will be on each section for the four sub areas. It is hard to know, the exact contribution, pros and cons for each work. What is the benefit of knowing their methods being numerical, or experimental, or taking any specific models? What would be the nexgt step?
We thank reviewer for the valuable comments on the paper. The papers are chose from those that are written in English from 2014 to 2022.each subtitle consists of brief of papers that method, propose and results are discussed.
Also next steps are written in in conclusion part.according to this work researchers of pavement engineering would know the way of their work in fields of construction,performance,maintenance and cost.so it would help them to choose the best model.
editing of English language was done by a native speaker according to reviewer’s comments.

Round 2
Reviewer 2 Report
The Tables and Figures are still not enumerared correctly. Table 2 is not mentioned in the text, at least I did not find it. Figures are referred as numbers from Fig.3-8, but there are only 6 Figures.
The answer about the time-slot for the review report is not satisfactorily explained. What is the meaning of "The first application and basis of NN was in the previous paper that the dear editor advised to omit it."? For example, there is one paper from 2009, using counterpropagation neural network models: Journal of chemometrics. 2009, vol. 23, issue 6, 283-293 (https://doi.org/10.1002/cem.1229)
Author Response
The authors would like to thank the area editor and the reviewers for their precious time and
invaluable comments. We have carefully addressed all the comments. The corresponding changes and refinements made in the revised paper are summarized in our response below:
The Tables and Figures are still not enumerared correctly. Table 2 is not mentioned in the text, at least I did not find it. Figures are referred as numbers from Fig.3-8, but there are only 6 Figures.
The table 2 is mentioned in the text in page No6 according to reviewer’s comment.
Number of figures corrected from 1 to 6 in the text.
The answer about the time-slot for the review report is not satisfactorily explained. What is the meaning of "The first application and basis of NN was in the previous paper that the dear editor advised to omit it."? For example, there is one paper from 2009, using counterpropagation neural network models: Journal of chemometrics. 2009, vol. 23, issue 6, 283-293 (https://doi.org/10.1002/cem.1229)
The authors meant that in the previous article that was reviewed, the basic application of neural network with images was given, which the editor advised to delete because it is a basic discussion

Reviewer 3 Report
This work has contributed to the body of knowledge in the area of AI for civil engineering, with a focus on pavement prediction task. While I find the work is interesting, some revisions will be requested:
1. It remains unclear about the utilised scholar databases, with other details, in section 2.
2. What are the research questions for this work?
3. While section 2 or 3 is for AI in flexible pavement, what is the taxonomy of the further classified tasks in this area? I would expect an insightful and meaningful summarization for the readers, rather than piling up the information for each paper.
4. What is the role of Fig. 2?
5. Table 2 is ugly, it will need improvment.
6. It is the same for table 3. Not sure about the findings.
Author Response
The authors would like to thank the area editor and the reviewers for their precious time and
invaluable comments. We have carefully addressed all the comments. The corresponding changes and refinements made in the revised paper are summarized in our response below:
This work has contributed to the body of knowledge in the area of AI for civil engineering, with a focus on pavement prediction task. While I find the work is interesting, some revisions will be requested:
- It remains unclear about the utilised scholar databases, with other details, in section 2.
The database that are used in the paper are shown in Fig 6
- What are the research questions for this work?
What are the practical algorithms for pavement engineering in fields of performance, maintenance, construction and cost?
- While section 2 or 3 is for AI in flexible pavement, what is the taxonomy of the further classified tasks in this area? I would expect an insightful and meaningful summarization for the readers, rather than piling up the information for each paper.
The taxonomy was done in 4 fields that any paper was shown in a paragraph with a summary and conclusion. And also in discussion and conclusion part the summarization and comparsion of models and discussed fields were done and suggestions for future works were presented
- What is the role of Fig. 2?
It is an example of DT structurte
- Table 2 is ugly, it will need improvement.
Table 2 was consist of Reference, Numerical and Experimental columns in initial paper that Methodology column was added by one of the reviewer’s comment.
- It is the same for table 3. Not sure about the findings.
The findings are correct and summarized from the papers.

Round 3
Reviewer 2 Report
OK
Author Response
The authors would like to thank the area editor and the reviewers for their precious time and invaluable comments

Reviewer 3 Report
1. Please justify and summarise the knowledge for each topic, rather than purely listing them. This requires a dedicated synthesis of the papers, which will be reflected in Table 2 and Table 3.
2. Please remove Fig 2, which is actually very annoying in the paper. Only DT is presented in Fig 2, while many other algorithms are available. It will needs more discussion and focus.
Author Response
The authors would like to thank the area editor and the reviewers for their precious time and invaluable comments. We have carefully addressed all the comments. The corresponding changes and refinements made in the revised paper are summarized in our response below:
- Please justify and summarise the knowledge for each topic, rather than purely listing them. This requires a dedicated synthesis of the papers, which will be reflected in Table 2 and Table 3.
We thank the reviewer for precious observations. Summary part added to table 2 and table 3.
- Please remove Fig 2, which is actually very annoying in the paper. Only DT is presented in Fig 2, while many other algorithms are available. It will needs more discussion and focus.
Fig 2 removed according to reviewer’s valuable comment and number of figures corrected from 1 to 5.
